# Proteomic Studies of the Biofilm Matrix including Outer Membrane Vesicles of *Burkholderia multivorans* C1576, a Strain of Clinical Importance for Cystic Fibrosis

**DOI:** 10.3390/microorganisms8111826

**Published:** 2020-11-19

**Authors:** Lucrecia C. Terán, Marco Distefano, Barbara Bellich, Sara Petrosino, Paolo Bertoncin, Paola Cescutti, Daniele Sblattero

**Affiliations:** Department of Life Sciences, University of Trieste, 34127 Trieste, Italy; lucreteran@gmail.com (L.C.T.); marco.diste86@gmail.com (M.D.); bbellich@units.it (B.B.); sar.petrosino@gmail.com (S.P.); pbertoncin@units.it (P.B.); pcescutti@units.it (P.C.)

**Keywords:** *Burkholderia multivorans*, proteomic, biofilm, outer membrane vesicles, LC-LC/MS

## Abstract

Biofilms are aggregates of microbial cells encased in a highly hydrated matrix made up of self-produced extracellular polymeric substances (EPS) which consist of polysaccharides, proteins, nucleic acids, and lipids. While biofilm matrix polysaccharides are unraveled, there is still poor knowledge about the identity and function of matrix-associated proteins. With this work, we performed a comprehensive proteomic approach to disclose the identity of proteins associated with the matrix of biofilm-growing *Burkholderia multivorans* C1576 reference strain, a cystic fibrosis clinical isolate. Transmission electron microscopy showed that *B. multivorans* C1576 also releases outer membrane vesicles (OMVs) in the biofilm matrix, as already demonstrated for other Gram-negative species. The proteomic analysis revealed that cytoplasmic and membrane-bound proteins are widely represented in the matrix, while OMVs are highly enriched in outer membrane proteins and siderophores. Our data suggest that cell lysis and OMVs production are the most important sources of proteins for the *B. multivorans* C1576 biofilm matrix. Of note, some of the identified proteins are lytic enzymes, siderophores, and proteins involved in reactive oxygen species (ROS) scavenging. These proteins might help *B. multivorans* C1576 in host tissue invasion and defense towards immune system assaults.

## 1. Introduction

In natural settings and in man-made environments, many bacteria are generally found to live in highly complex communities referred to as biofilms, which are surface-associated aggregates of microbial cells embedded in a matrix consisting of extracellular polymeric substances (EPS) [1,2,3,4]. Although the composition of the biofilm matrix varies with the bacterial species and the growth conditions under which biofilms develop, EPS generally include polysaccharides, proteins, nucleic acids, and lipids [5]. The biofilm lifestyle allows bacterial cells to experience a physical closeness which is the base for phenomena such as quorum sensing (QS) and horizontal gene transfer (HGT) [6]. Moreover, the biofilm matrix provides bacteria with protection towards dehydration, UV light, disinfectants, and toxic metal ions, some antimicrobial compounds, many protozoa, and attacks by the immune system of the host [6]. As a generality, the composition of the biofilm matrix consists of exopolysaccharides, proteins, and nucleic acids, although the components and their proportions are dependent on the microorganism and the conditions of their growth [5]. In floating activated sludge, the protein content can even exceed that of polysaccharides [7,8,9]. Matrix-associated proteins are represented by membrane surface adhesins, proteins building up extracellular appendages (flagella, pili, and fimbriae), proteins actively secreted, and proteins transported by outer membrane vesicles (OMVs) [5].

OMVs are recognized as biofilm matrix components [10] and their biological role has been related to protective mechanisms [11], as they are involved in a wide range of phenomena like pathogenesis, bacterial communication, bacteria-host interactions, nutrient capture, HGT, and competition [12,13]. OMVs are bi-layered structures with a diameter ranging from 20 to 200 nm, which gemmate from the outer membrane (OM) of many Gram-negative bacteria [14,15]. OMVs generally carry OM lipids, lipopolysaccharide, proteins, and nucleic acids [16]. 

*Burkholderia multivorans* is a Gram-negative opportunistic pathogen that can cause severe lung infections in cystic fibrosis (CF) patients [17]. It belongs to the so-called *Burkholderia cepacia* complex (Bcc), a group of at least 22 closely-related species widely distributed in natural environments and with the ability to infect plants, animals, and humans [18,19,20,21,22]. When grown in Petri dishes containing different solid culture media, Bcc species have been shown to produce various biofilm polysaccharides or exopolysaccharide (Epols), most often in mixtures [23]. Furthermore, the specific biofilm mode of growth may influence the production of distinct Epols [24].

In this work, we used *B. multivorans* strain C1576, a CF clinical isolate biofilm-growing species which was thoroughly investigated for Epol production in our laboratory. It produces cepacian, the most common Epol among both Bcc and non-Bcc species when grown on semipermeable nitrocellulose membranes deposited on solid yeast extract–mannitol culture medium [25]. When grown in the same way but using Mueller–Hinton medium, it synthesizes a different exopolysaccharide which has been named EpolC1576. The latter consists of a tetrasaccharide repeating unit containing equimolar amounts of D-mannose and D-rhamnose [25]; this polymer might be involved not only in biofilm formation and maintenance, but also in the diffusion of small nonpolar compounds like quorum sensing molecules, through the highly hydrated environment of the biofilm matrix [26].

The presence of proteins has been recognized in biofilms of many bacterial species, but their identification and functional characterization have been addressed only for a limited number of bacteria [27], including some *Burkholderia* species [28,29,30]. At the same time, *Burkholderia* OMVs have been investigated mainly in the frame of vaccine development [31,32,33], while there is little information about their proteome [34]. Therefore, the aim of this work was to elucidate the proteins associated with the biofilm matrix of *B. multivorans* C1576.

## 2. Materials and Methods

### 2.1. Bacterial Strain and Culture Conditions

*B. multivorans* C1576 (LMG 16660), a reference strain that belongs to the panel of *B. cepacia* complex strains (EP1), was purchased from the BCCM^TM^ bacteria collection. The biofilm was developed on cellulose membranes (Sigma, cut-off 12.400 Da) [35] that were prepared as follows: they were cut in circles to fit the Petri dish, boiled in 5% Na_2_CO_3_ solution for 15 min and then boiled in water for 15 min, subjected to autoclave sterilization, and laid on Petri dishes containing MH agar medium [36]. Membranes were extended all over the plate and the surplus of water was let evaporate under the hood. An overnight liquid culture of *B. multivorans* C1576 grown in MH medium was diluted to obtain a cell suspension having 0.13 OD at 600 nm (about 1 x10^6^ CFU/mL), three aliquots of 10 µL of the cell dilution were transferred on the membranes deposited on the Petri dishes, and they were incubated at 30 °C for 7 days.

### 2.2. OMV Isolation and Proteins Preparations

After 7 days of growth, the biofilm formed by *B. multivorans* C1576 was recovered with sterile NaCl 0.9% (7 mL for each membrane) by scraping it from the membranes with a sterile spatula. During harvesting procedures, the cell suspension was kept on ice. To reduce contamination risks with cellular proteins, the cell suspension was vortexed for 2 min, avoiding chemical treatment and sonication for biofilm disruption. The cell suspension was then centrifuged at 24,000× *g* for 30 min at 4 °C (Beckman J2-21 M/E Centrifuge), the supernatant was filtered first, through a 0.45-µm pore size filter (Millex-HA Non-Sterile Syringe Filter) and then through a 0.22-µm pore size filter (KX Sterile Syringe Filter, Kinesis) to eliminate residual bacterial cells from the supernatant. Half of the filtered supernatant was frozen at −20 °C, whereas the other half filtered supernatant was ultra-centrifuged at 100,000× *g* for 3 h at 4 °C to pellet OMVs.

After centrifugation, the supernatant was considered for analysis as the “matrix” sample, while the pellet was resuspended with water and considered as the “OMVs” sample. Both samples were split in two aliquots, one stored at 4 °C and the other one at −20 °C.

### 2.3. Transmission Electron Microscopy (TEM) Imaging of OMVs

An aliquot of the stock solution and a 1:50 dilution of the pellet obtained after ultracentrifugation of the biofilm matrix of *B. multivorans* C1576 were prepared for TEM imaging using the following procedure: a droplet of the sample was deposited on a carbon-coated copper grid (Electron Microscopy Science, CF 200-Cu Carbon film) to allow sample adsorption and then diluted with a droplet of water. Later, the sample was negatively stained by adding a droplet of 2% uranyl acetate. After the negative staining step (2–5 min), liquid excess was removed with filter paper and the sample was air-dried. Sample images were acquired with an EM208 TEM (Philips, Netherlands) equipped with a Quemesa camera (Olympus Soft Imaging Solutions) using Radius software (EMSIS, Münster, Germany).

### 2.4. Protein Preparations for Liquid Chromatography coupled to Tandem Mass Spectrometry (LC-MS/MS)

Protein fraction of both *B. multivorans* C1576 matrix and OMVs were concentrated by using a Millipore Microcon centrifugal filter device with a 3 kDa molecular weight cut-off. Samples were separated by SDS-PAGE using a 12% Tris Glycine polyacrylamide. Stock solutions were diluted 1:2 and 1:5, and 15 µL of both samples were mixed with 6x SDS loading buffer (0.25 M Tris-HCl pH 6.8, 8% SDS, 40% sucrose, 0.5% β-Mercaptoethanol, 0.2% bromophenol blue) and boiled for 5 min. The gel was electrophoresed at 80 V for 5 min and then at 120 V for 60 min in Tris-Glycine SDS buffer and stained with Coomassie Blue.

Subsequently, electrophoretic lanes were cut into five equivalent parts, and proteins were processed as previously described [37].

### 2.5. Protein Identification

Peptide masses and MS/MS spectra were exported as ‘.mgf’ files and database search was performed with the MASCOT MS/MS Ion Search option (www.matrixscience.com) with the following set up parameters: fixed modification: none; variable modification: none; peptide tol.: 1.2 Da; MS/MS tol.: 0.6 Da; peptide charge: 2 and 3; mass: monoisotopic; instrument: ESI-TRAP. On average, individual ions scores > 68 indicate identity or extensive homology (*p* < 0.05). The relative abundance was calculated according to the most frequent occurrence and higher scores during the LC-LC/MS identification.

### 2.6. Bioinformatic Analysis

All the identified proteins were classified following two different criteria. On one hand, the classification was done accordingly to the cluster of orthologous group (COG) categories with COGnitor [38] run by Operon Mapper [39] and on the other hand, they were classified according to the subcellular localization using the web-based software PsortDB version 3.0 [40]. Both classifications were performed for each of the identified proteins as well as for all the proteins that are codified in the genome of strain C1576 of *B. multivorans* (Genbank Accession Number: JAAKGD000000000.1). The identified proteins were searched in Uniprot Database for considering parameters such as isoelectric point and molecular weight.

The hypergeometric distribution was assayed to evaluate the enrichment of the different protein categories (in terms of both function and subcellular localization) encoded in the genome of *B. multivorans* C1576 compared to the ones identified in the matrix and in the OMVs. Interaction networks were constructed with STRING v11 [41] for the proteins identified in the Biofilm and in the OMVs using a moderate confidence level of 0.4. All types of protein–protein interactions present in STRING were selected for the analysis, including known interactions (from curated databases and experimentally determined), predicted interactions (gene neighborhood, fusions, and co-occurrence), and others interactions (text mining, coexpression, and protein homology). Additionally, the expected interactions between random proteins were estimated through STRING v11 [41]; comparing the number of edges of random proteins with the ones obtained in our set of proteins is meaningful since proteins that have more interactions than expected for a random set of proteins indicates that they are biologically related.

## 3. Results

### 3.1. Strategy for Biofilm Production and Protein Separation

*B. multivorans* C1576, a reference strain that belongs to the panel of *B. cepacia* complex strains (EP1), was selected for the identification of biofilm matrix-associated proteins. The general work scheme is shown in Figure 1. The protocol used a colony-based biofilm system in which *B. multivorans* C1576 was seeded on semipermeable nitrocellulose membranes deposited on solid Mueller Hinton plates [25] and grown for 7 days. The biofilm was recovered from nitrocellulose, separated from bacterial cell pellet, and finally divided by ultracentrifugation in 2 fractions identified as OMVs and matrix. Both fractions were then initially analyzed via 1D SDS-PAGE further coupled to Liquid Chromatography Tandem-Mass Spectrometry (LC-MS/MS) to disclose the identity of proteins of biofilm matrix samples.

### 3.2. Visualization of OMVs Produced by Biofilm-Growing B. multivorans C1576

The *B. multivorans* C1576 capability to produce OMVs was probed through observation via transmission electron microscopy (TEM) of the resuspended pellet obtained after ultracentrifugation of the purified biofilm matrix. Many bi-layered spherical formations were observed (Figure 2) together with elongated structures that might represent contamination with bacterial appendages (flagella, pili, and/or fimbriae). The majority of OMVs released by *B. multivorans* C1576 showed diameters that ranged from 25 to 70 nm, while OMVs with a diameter exceeding 150 nm were only occasionally observed. The presence of OMV-like structures in the matrix of *B. multivorans* C1576 reinforces the hypothesis that proteins associated with OMVs are an integral part of the biofilm matrix proteome.

### 3.3. Identification of B. multivorans C1576 Proteins Associated with the Matrix and OMVs

The 1D SDS-PAGE analysis of OMV and matrix fractions (Figure 3A) revealed a different protein profile among the two samples. Both samples were subjected to mass spectrometry analysis and a total of 161 different proteins were found associated with the biofilm matrix (Appendix A), whereas 64 proteins were identified from the OMV sample (Appendix A). Out of these proteins, the twenty most abundant ones for both biofilm matrix and OMVs are listed in Table 1 and Table 2, respectively. When comparing the two proteomic datasets of *B. multivorans* C1576, it was interesting to notice that 29% of OMV-associated proteins (19 out of 64) were in common with the biofilm matrix proteome (Figure 3B and Table 3). On the other hand, when considering the twenty most abundant matrix-associated proteins, only five of these were shared with the ones found in OMVs: an acyl carrier protein (*acpP*—ACP_BURCA), a beta-ketoacyl-ACP reductase (Bmul_2181—AOL03371.1), a thiol: disulfide interchange protein DsbA/DsbL (*dsbA*—WP_006398112.1), a thioredoxin (Bmul_1445—ABC39250.1), and an endoribonuclease (Bmul_0927- WP_006398680.1), thus suggesting that OMVs do not represent an important source of proteins for the biofilm matrix of *B. multivorans* C1576.

### 3.4. Functional Classification of the Identified Proteins from the Biofilm Matrix and OMVs

To determine the function of the identified proteins, they were classified into functional COG categories. These functional categories group proteins according to their function and are represented by letters. After classification of the 161 different proteins identified in the biofilm matrix, the category with the highest percentage of members, corresponding to 30 different proteins (19% of the total proteins) was the “energy production and conversion” one, represented by the letter C (Figure 4A). Other categories with a high frequency of protein members identified were the ones related to “metabolism and transport of amino acids” (letter E) and “posttranslational modifications” (letter O), represented by 21 and 19 proteins, respectively. Lower percentages were for the categories “translation” (J, 8%), “metabolism and transport of nucleotides” (Q, 6%), “lipid metabolism” (I, 5%), and “metabolism and transport of carbohydrates” (G, 5%). Finally, 19% of the proteins belonged to poorly characterized proteins and were distributed in the general prediction function (R) and unknown function (S) categories.

To study if there was an enrichment of some of the functional categories to which the identified proteins belonged, a hypergeometric distribution was performed: in this way, we could compare the identified proteins with the ones encoded in the whole genome of *B. multivorans* C1576. As shown in Figure 4B and Appendix A, the enriched categories were related with C, J, O, Q, and S, while the underrepresented categories corresponded to proteins belonging to the categories “cell wall structure, biogenesis, and outer membrane” (M), “transcription” (K), “transport of inorganic ions” (P), and “unknown function” (R).

When analyzing the 64 identified proteins of the OMV proteome, different results were obtained. In this set, the higher percentage of proteins identified were belonging to the categories of “unknown function” (S, 14%), “cell wall structure, biogenesis, and outer membrane” (M, 12%) transport and metabolism of aminoacids (E, 11%), production and conversion of energy (C, 9%), post-translational modifications (O, 5%), and transport of inorganic ions (P, 5%). When compared to the whole genome of *B. multivorans* C1576, as it is shown in Figure 4B, the enriched categories correspond to S, Q, M, O, and C, but the difference between enriched and underrepresented categories is narrower for the OMVs than for the identified matrix proteins. Noteworthy, the categories “transcription” (K) and “metabolism and transport of nucleotides” (F) categories were only represented in the identified proteins from the matrix, but not from OMVs.

### 3.5. Subcellular Localization of the Identified Proteins from the Biofilm Matrix and OMVs

In addition to their function, the identified proteins were classified according to the subcellular localization. Out of the 161 proteins of the matrix proteome, 90 were predicted to be cytoplasmic, 26 to be periplasmic, 3 to be extracellular, 2 to be located in the cytoplasmic membrane, and 1 in the outer membrane; the remaining 38 were of unknown subcellular localization (Figure 5A). According to the hypergeometric distribution proteins belonging to the subcellular localization of the cytoplasmic and periplasmic were enriched, those located in the cytoplasmic membrane were reduced, while the remaining categories cytoplasmic membrane, outer membrane, extracellular, and unknown were found as expected (Figure 5B and Appendix A).

The 64 proteins of the OMVs were also mainly represented by proteins predicted to be of cytoplasmic localization (29 proteins), while 13 proteins were of outer membrane origin and 8 periplasmic. The 14 proteins left were predicted as unknown localization. For this set of proteins, those from the outer membrane and periplasmic origin were enriched as their relative proportion was higher than among the proteins encoded by the whole genome of *B. multivorans* C1576.

### 3.6. Protein–Protein Interaction Networks of the Identified Proteins from the Biofilm Matrix and OMVs

Finally, we investigated the possible interactions among the identified proteins. Protein–protein interactive networks for the biofilm matrix as well as the OMVs of *B. multivorans* C1576 were found after applying the STRING v11 database. In each of the networks, proteins are represented with nodes and interactions with edges; the strongest the interaction, the thickest the line. As shown in Figure 6, 117 out of the 161 proteins identified in the biofilm matrix interact with each other through 285 edges with different strengths; it is calculated by STRING v11.0 database that random proteins would have 133 edges, these data indicate that the network has more than double the interactions expected (133 edges), suggesting a biological connection between these proteins. The remaining 44 proteins did not show any interaction. Eighteen proteins interacting with each other belong to the energy production and conversion category with 44 edges, in line with the data that this category is one of the enriched ones. Moreover, transcription and posttranslational modifications are enriched categories represented by 7 and 11 nodes, respectively, with strong interactions. Meanwhile, the category metabolism and transport of amino acids is represented by 8 proteins as expected in the hypergeometric aspect (Figure 5B).

Regarding the OMVs’ proteins, 24 out of the 64 identified proteins show no interactions within each other, while the remaining 40 proteins were predicted to exhibit interactions with each other through 52 edges. In addition, in this case, the network has more interactions than expected (21 edges) for a random set of proteins, strongly indicating that this group of proteins has a related biological role. The functional category of cell wall structure, biogenesis, and outer membrane (M) is represented by 5 nodes, which is also in accordance with the enrichment of this category in the OMVs, while in the biofilm matrix this category is impoverished. Additionally, the categories’ posttranslational modifications (O) and lipid metabolism (I) are enriched and represented in the network (Figure 7). This network supports the interaction of proteins related to the enriched categories in agreement with the hypergeometric distribution test.

## 4. Discussion

Even though extracellular proteins have been proposed to be of primary importance for the function of the biofilm matrix, there is still a lack of knowledge about the identity of many of them. In this work, matrix-associated proteins of biofilm-growing *B. multivorans* C1576 have been identified. It has not been surprising to find that a large fraction of the matrix proteome of *B. multivorans* C1576 is made up of membrane-bound and cytoplasmic proteins. Membrane and cytoplasmic proteins have been recognized in the biofilm matrix of various bacterial species [42,43,44,45], even in those characterizing microbial communities of activated sludges and acid mine drainage [46,47]. Despite this, the reason why these proteins localize in the biofilm matrix is still elusive.

Biofilm production involves bacterial cell death, an event that is thought to be characteristic of developing biofilms [42], thus suggesting that cell lysis could explain the presence of membrane-bound and cytoplasmic proteins in the matrix. Another source of intracellular proteins for the matrix could be represented by spherical bilayered structures from the outer membrane, the OMVs, that planktonic and biofilm-growing Gram-negative bacteria have been shown to release [12]. *Burkholderia thailandensis* OMVs range in size from 20 to 100 nm and contain antimicrobials [34], while *Burkholderia cepacia* releases OMVs that range from 30–220 nm in diameter and contain virulence factors [48]. Similarly, our results showed that most of the OMVs of *B. multivorans* C1576 have a diameter that ranges between 25 and 70 nm and some of them reached 150 nm (Figure 2).

These spherical bilayered structures that gemmate from the outer membrane have been recognized to be a relevant component of the biofilm matrix of other Gram-negative bacteria such as *P. aeruginosa* [12]. The proteome of OMVs produced by biofilm-growing *B. multivorans* C1576 consisted of 64 proteins. Many of them are outer membrane proteins and this is in good agreement with previous findings [16,49]. OMVs contain porins, siderophores, and enzymes scavenging reactive oxygen species. Proteomic data reveal that about 29% of OMV-associated proteins (19 out of 64) are in common with the matrix proteome (Figure 3), thus indicating that OMVs could represent a source of both membrane-bound and cytoplasmic proteins for the biofilm matrix of *B. multivorans* C1576. As observed for other Gram-negative species [49,50], OMVs are greatly enriched in outer membrane proteins including porins (*opcP*; Bmul_4327; Bmul_4600), receptors (Bmul_1594; Bmul_3338; Bmul_4173) and lipoproteins (*slyB*). However, only 5 proteins out of the 20 most abundant matrix-associated proteins are found in OMVs.

Our data suggest that cell death and release of OMVs are mainly responsible for the composition of the extracellular proteome of biofilm-growing *B. multivorans* C1576. It would be interesting to know whether some of these proteins possess enzymatic activity and/or a structural role that is crucial for the biofilm physiology. Not surprisingly, various lytic proteins that could be involved in the enzymatic degradation of different molecules have been identified, such as the peptidase S1 (Bmul_3442), the proteolytic subunit of the ATP-dependent Clp protease (Bmul_1349—*clpP*), and an endoribonuclease (Bmul_1781). Even though some of these proteins have not been characterized as virulence factors yet, it cannot be ruled out that they could support *B. multivorans* C1576 infections via the degradation of host tissues. These enzymes could also help *B. multivorans* C1576 to resist attacks perpetrated by phagocytic cells and other microorganisms.

It is interesting to note that a bacteriocin (Bmul_5413—***lin***) has also been found. Bacteriocins are powerful toxins with an antibacterial activity that mediate competition between different bacterial species [51]. Since *P. aeruginosa* and Bcc species can co-infect CF patients [52], it can be speculated that bacteriocins could help *B. multivorans* C1576 in shaping and defining the relationship with *P. aeruginosa* within CF lungs.

Among the matrix-associated proteins, the well-characterized virulence factor named ecotin (***ecoT***-WP_006405712.1) has been detected. The one produced by *P. aeruginosa* has been shown to inhibit the serine protease elastase synthesized by neutrophils [53], thus severely compromising the bactericidal efficiency of this enzyme. In the same way, ecotin produced by *B*. *multivorans* C1576 can protect the bacteria from killing activity of elastase secreted by neutrophils, thus enhancing bacterial tolerance towards the innate immune system.

Proteins responsible for detoxification of reactive oxygen species (ROS) are a relevant fraction of both matrix- and OMVs-associated proteins. Proteins found to be involved in oxidative stress are four thiol peroxidases *tpx* (SMG02352.1, WP_006400399.1, WP_057926826.1, and WP_059465341.1), a superoxide dismutase *sod* (EGD01013.1), catalase HPII (*katE*, CATE_ECOLI), a peroxidase (Bmul_2571, OJD07310.1), and a thioredoxin (Bmul_1445, ABC39250.1). Since some of these detoxifying enzymes have been found associated with OMVs, it may be speculated that *B. multivorans* C1576 releases them via OMVs to cope with oxidative stresses. The presence of matrix-associated enzymes that inactivate ROS would allow bacteria to resist to those attacks from phagocytes based on hydrogen peroxide production [54].

It has been interesting to find that OMVs carry various proteins involved in iron acquisition, also referred to as siderophores. Acquisition of iron is crucial for bacterial survival and siderophores have been recognized as relevant virulence factors for many pathogens affecting CF patients [55,56]. Since, in body fluids iron is tightly bound to proteins such as transferrin, lactoferrin, and ferritin [57], bacterial pathogens have developed strategies to utilize these proteins as sources of iron. For example, in CF lungs *B. cenocepacia* can acquire iron after proteolytic degradation of ferritin [58]. In the same way, siderophores identified in *B. multivorans* C1576 could help this opportunistic pathogen to survive within the host environment where availability of free iron is strictly limited.

The biological role of the intracellular proteins associated with both the biofilm matrix and OMVs of *B. multivorans* C1576 is not known. There is experimental evidence that some intracellular proteins may have a function also outside the cell and for this reason they have been named moonlighting proteins [59]. Among these proteins, there is the elongation factor Tu which has been localized on the surface of *P. aeruginosa* where it helps bacteria to evade the human complement attack via interactions with host regulatory proteins [60]. The matrix proteome of *B. multivorans* C1576 has been found to contain various elongation factors (G, Ts, GreAB), but not the elongation factor Tu. The latter, together with chaperonin GroEL, has been also shown to enable lactobacilli to bind to mucin and epithelial cells of the human gastrointestinal tract [61,62]. Since some of these intracellular proteins have been demonstrated to be involved in host-pathogen interactions for other bacterial species, it can be speculated that one or more intracellular proteins localized in the biofilm matrix of *B. multivorans* C1576 could play a role in bacterial pathogenesis. Moreover, the presence of the mentioned chaperons and elongation factors, plus ribosomal proteins (A9ADI4) of cytoplasmic origin in OMVs was also described for *E. coli*, suggesting that, as the translation of outer membrane proteins occurs simultaneously with the integration into the membrane, transcriptional and ribosomal proteins can be included into OMVs [49,63,64].

Many of the identified proteins are membrane-bound proteins and of cytoplasm origin, thus suggesting that cell lysis could represent a relevant source of proteins for the biofilm matrix. They are porins, ATP binding-cassette transporters (ABC), enzymes of the tricarboxylic acid cycles (TCA), proteins involved in fatty acids biosynthesis, cell redox homeostasis, and translation. Moreover, the presence of the bacterial DNA-binding proteins indicates that DNA molecules are components of the biofilm matrix of *B. multivorans* C1576, as already seen for other bacterial species [65,66,67,68,69].

Extracellular DNA molecules (eDNA) have been recognized as a prominent constituent of the bacterial biofilm matrix [65,70]. For this reason, it has not been unexpected to find DNA-binding proteins in the proteome of the biofilm matrix of *B. multivorans* C1576. The DNA-binding protein HU-alpha *hupA* (DBHA_BURPS), which is a histone-like protein involved in DNA wrapping, has been found associated with OMVs, whereas the DNA-binding protein Bv3F (Bmul_0190, POM19719.1) and a single-stranded DNA-binding protein (Bmul_0538, SMG01792.1) have been localized in the biofilm matrix. Even though DNA-binding proteins affect DNA compactness and transcription [71], it is unknown whether they influence the structural and functional properties of eDNA in the context of the biofilm matrix.

Beyond exoenzymes, membrane-bound and DNA-binding proteins, the biofilm matrix of bacteria also contains lectins, proteins with no enzymatic activity, but showing specific carbohydrate-binding ability. Lectins are thought to take part in biofilm formation and development, but also to be responsible for its architecture and mechanical stability. Transcriptomic and proteomic analysis of mutants of *B. cenocepacia* H111 led to the identification of the quorum sensing-regulated operon *bclACB* which codes three lectins: BclA, BclC, and BclB [72]. *B. cenocepacia* H111-*bclACB* mutants produce biofilms with altered architecture with respect to the wild-type, thus suggesting that these three lectins are necessary for biofilm proper development [72]. In the search for proteins of *B. multivorans* C1576 that may interact with EpolC1576, no known carbohydrate-binding protein has been found.

## 5. Conclusions

In conclusion, we have shown that biofilm-growing *B. multivorans* C1576 produces OMVs and that these spherical bilayered structures represent a source of proteins for the biofilm matrix together with cell lysis. Our results also elucidate the proteome from a functional perspective of the biofilm matrix and OMVs of *B. multivorans* C1576. This work opens perspectives to disclose the role played by the OMVs within biofilms produced by *B. multivorans* C1576.

## Figures and Tables

**Figure 1 microorganisms-08-01826-f001:**
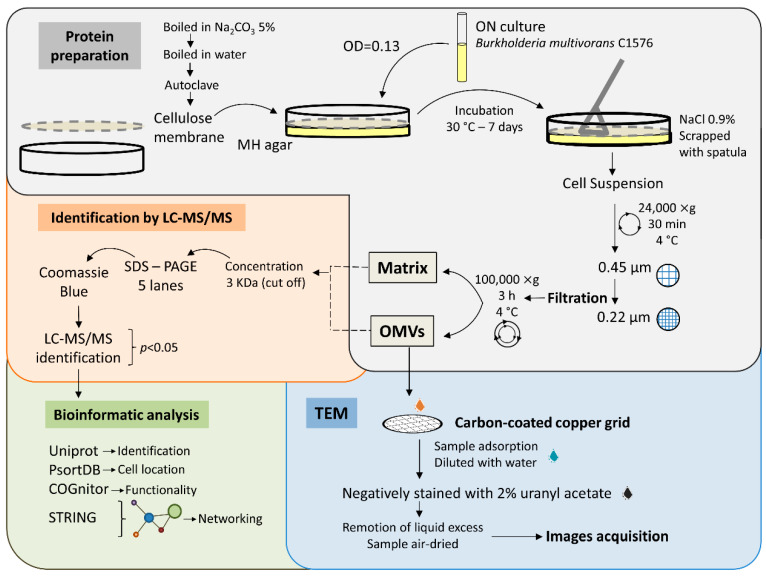
The general work scheme followed in the present work. Proteins from biofilm were separated into 2 fractions called Matrix and outer membrane vesicles (OMVs); both fractions were subjected to identification by LC-MS/MS and bioinformatics analysis. OMVs’ fraction was further analyzed by transmission electron microscope (TEM).

**Figure 2 microorganisms-08-01826-f002:**
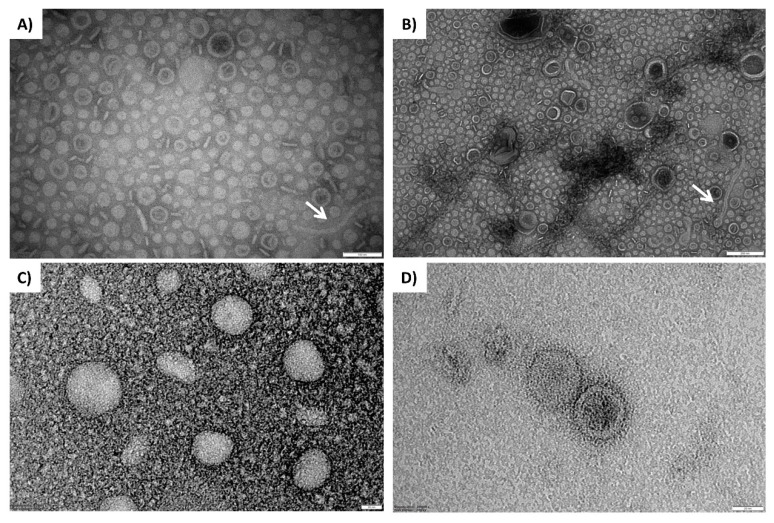
TEM visualization of OMVs produced by biofilm-growing *B. multivorans* C1576. The upper micrographs show TEM imaging of OMVs of the stock sample; elongated structures could represent contaminations with bacterial appendages (white arrows). The lower micrographs show TEM imaging of OMVs coming from the 1:50 dilution of the stock sample. Scale bars: 100 nm (**A**), 200 nm (**B**), 20 nm (**C**), and 20 nm (**D**).

**Figure 3 microorganisms-08-01826-f003:**
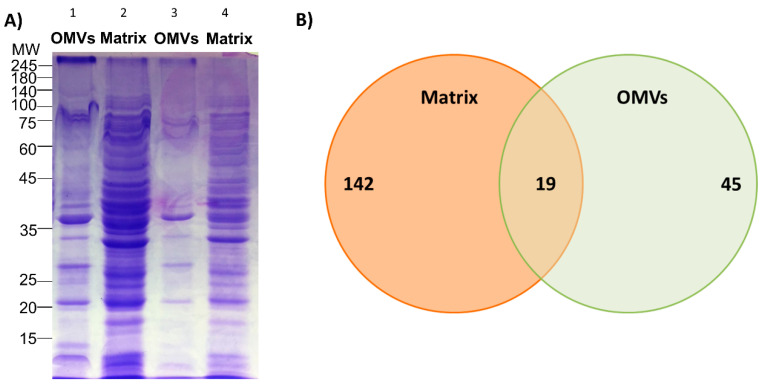
(**A**) 12% SDS-PAGE protein profile of OMVs and the biofilm matrix produced by *B. multivorans* C1576; 15 µL of both samples were loaded at 1:2 (lane 1 and 2) and 1:5 (lane 3 and 4) dilution from stock concentration. (**B**) Venn diagram showing proteins of the biofilm matrix, the OMVs, and the ones that overlap between both samples.

**Figure 4 microorganisms-08-01826-f004:**
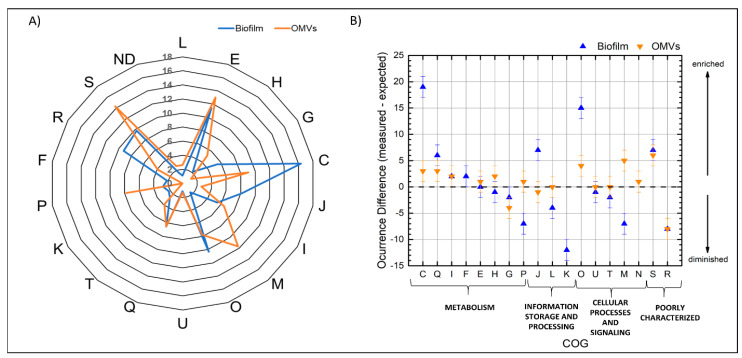
(**A**) Percentage of the cluster of orthologous group (COG) functional categories of the identified proteins from the biofilm matrix and the OMVs. (**B**) Difference between the measured and the expected occurrence of the hypergeometric analysis. The positives values represent the enrichment of certain categories while the decrease of certain functional categories is represented by negatives values; when the expected value is identical to the measured one it is represented as zero. Letters represent the functional categories as follows: L: DNA replication, recombination and repair; E: metabolism and transport of amino acids; H: metabolism and transport of coenzymes; G: metabolism and transport of carbohydrates; C: production and conversion of energy; J: transcription; I: lipidic metabolism; M: Cell wall structure, biogenesis and outer membrane; O: posttranslational modifications; U: Intracellular trafficking, secretion, and vesicular transport; T: mechanisms of signal transduction; Q: metabolism and transport of nucleotides; P: transport of inorganic ions; N: cell motility; K: translation including ribosomes biogenesis; F: metabolism and transport of nucleotides; R: general prediction function only; S: function unknown; ND: not determined.

**Figure 5 microorganisms-08-01826-f005:**
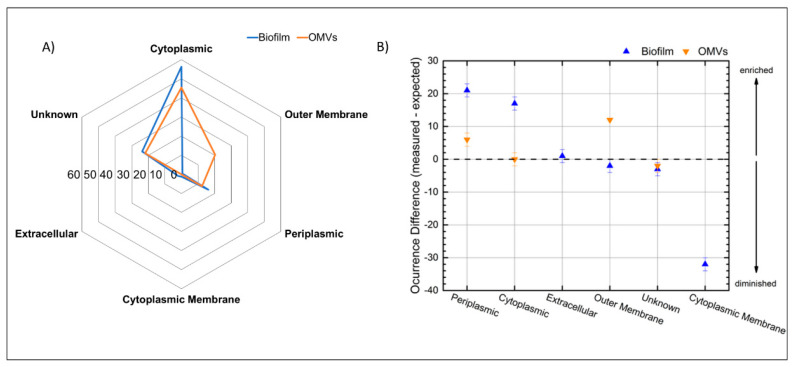
(**A**) Percentage of the subcellular localization of the identified proteins from the biofilm matrix and the OMVs. (**B**) Difference between measured and expected occurrence for the proteins of each subcellular localization. Zero represents no differences in occurrence and separates potentially enriched categories (positive values) from the potentially reduced categories (negative values).

**Figure 6 microorganisms-08-01826-f006:**
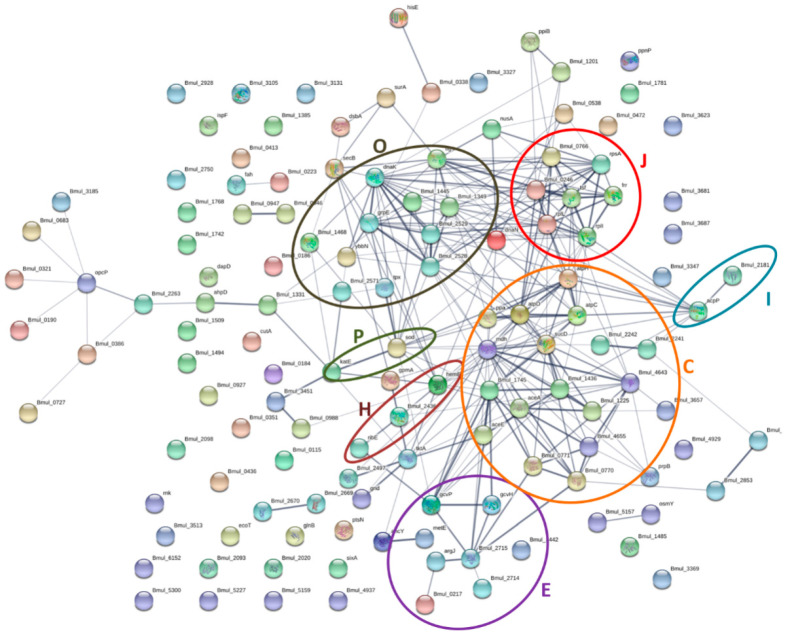
Protein–protein interaction network of the identified proteins of *B. multivorans* C1576 in the biofilm matrix. The circles highlight the proteins related with functional categories. C: energy production and conversion, J: translation, O: post-translational modification, E: transport and metabolism of aminoacids, P: transport and metabolism of inorganic ions, I: metabolism of lipids, and H: transport and metabolism of coenzymes. The proteins are represented by nodes, while the interactions among them are represented by edges. The strength of interactions is proportional to line thickness. The network was constructed with STRING v11.

**Figure 7 microorganisms-08-01826-f007:**
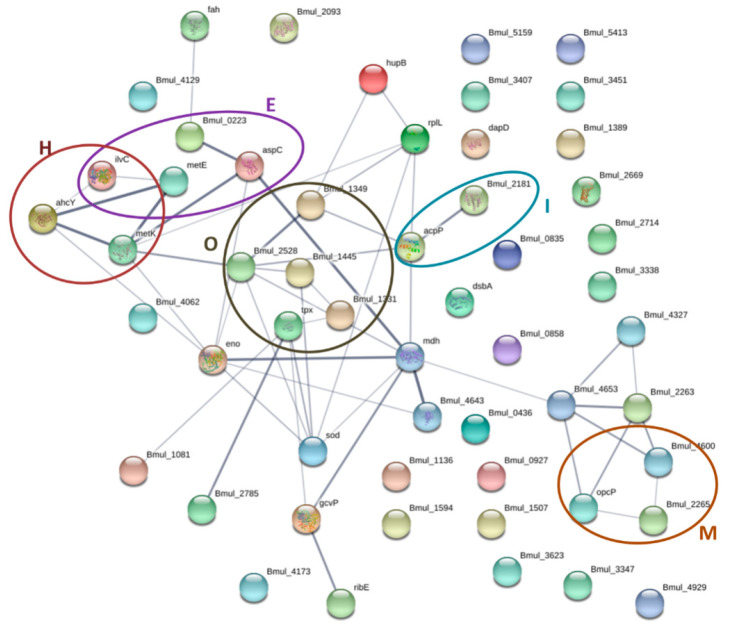
Protein–protein interaction network of the identified proteins of *B. multivorans* C1576 in the OMVs. The circles highlight the proteins related to functional categories. O: post-translational modification, E: transport and metabolism of aminoacids, I: metabolism of lipids, H: transport and metabolism of coenzymes, and M: cell wall structure, biogenesis and outer membrane. The proteins are represented by nodes while the interactions between them by edges. The strength of interactions is proportional to line thickness. The network was constructed with STRING v11.

**Table 1 microorganisms-08-01826-t001:** List of the twenty most prevalent matrix-associated proteins.

Proteins	Gene Name/Identifier	Accession Number ^a^	Number of Matches ^b^	Subcellular Localization ^c^
Acyl carrier protein	*acpP*	ACP_BURCA	35	cytoplasm
thiol peroxidase	*tpx*	WP_006400399.1	30	periplasmic space
carboxymuconolactone decarboxylase	*ahpD*	EGD01850.1	22	cytoplasm
beta-ketoacyl-ACP reductase	Bmul_2181	AOL03371.1	17	cytoplasm
branched-chain amino acid ABC transporter substrate-binding protein	*livK*/Bmul_3623	WP_006396439.1	16	periplasmic space
thiol: disulfide interchange protein DsbA/DsbL	*dsbA*	WP_006398112.1	15	periplasmic space
Malate dehydrogenase	*mdh*	WP_006399543.1	13	unknown
ABC transporter substrate-binding protein	Bmul_3347	WP_006397294.1	13	periplasmic space
Fe^(3+)^ ABC transporter substrate-binding protein	Bmul_2093	WP_035953191.1	11	periplasmic space
glutamate/aspartate ABC transporter substrate-binding protein	Bmul_2714	WP_006408338.1	10	periplasmic space
RNA chaperone Hfq	*hfq*/Bmul_1468	WP_063496600.1	10	cytoplasm
thioredoxin	Bmul_1445	ABC39250.1	9	cytoplasm
ribosomal protein S1	*rpsA*	EJO59460.1	9	cytoplasm
Protein-export protein SecB	*secB*	SECB_BURCA	8	cytoplasm
hypothetical protein	Bmul_2263	SAK23673.1	8	unknown
5-methyltetrahydropteroyltriglutamate-homocysteine S-methyltransferase	*metE*	WP_060098089.1	8	cytoplasm
NADP-dependent phosphogluconate dehydrogenase	*gnd*	WP_006399510.1	8	cytoplasm
Endoribonuclease L-PSP	Bmul_0927	WP_006398680.1	7	unknown
NADP-dependent isocitrate dehydrogenase	Bmul_0771	WP_006398523.1	7	cytoplsm
superoxide dismutase	*sod*	WP_006398503.1	6	periplasmic space

^a^ NCBI accession number of the proteins. ^b^ #matches: number of matching peptides during the identification. ^c^ Subcellular localization prediction for each of the proteins according to Psortdb v3.0.

**Table 2 microorganisms-08-01826-t002:** List of the top twenty of the most prevalent OMV-associated proteins.

OMV Proteins	Gene Name/Identifier	Accession Number ^a^	Number of Matches ^b^	Subcellular Localization ^c^
porin	*opcP*	WP_054315301.1	30	outer membrane
OmpA-like protein	Bmul_2265	WP_006400736.1	17	outer membrane
chaperonin GroEL	*groL*/Bmul_2528	WP_006400973.1	16	cytoplasm
porin	*opcP*	WP_059451674.1	15	outer membrane
iron complex outer membrane receptor protein	Bmul_1594	BAG43570.1	12	outer membrane
glutamate/aspartate ABC transporter, periplasmic glutamate/aspartate-binding protein	*gltI*	EED98255.1	10	unknown
hypothetical OmpA-like protein	*opcP*	WP_059451674.1	9	outer membrane
putative lipoprotein	Bmul_1389	WP_006402280.1	7	unknown
TonB-dependent hemoglobin/transferrin/lactoferrin family receptor	Bmul_3338	WP_048804300.1	7	outer membrane
aspartate/tyrosine/aromatic aminotransferase	*aspC*	WP_038714441.1	4	cytoplasm
adenosylhomocysteinase	*ahcY*	EEE03553.1	5	cytoplasm
acyl carrier protein	*acpP*	ACP_BURCA	4	cytoplasm
malate dehydrogenase	*mdh*	MDH_BURCJ	4	unknown
putative outer membrane protein	Bmul_1507	OJD05646.1	4	outer membrane
enolase	*eno*	ENO_BURM1	4	cytoplasm
aldehyde dehydrogenase family protein	Bmul_3451	WP_054317513.1	4	cytoplasm
citrate synthase	Bmul_4643	OJD04157.1	3	cytoplasm
redoxin family protein	Bmul_1331	KOS88719.1	3	cytoplasm
phasin protein	Bmul_1136	WP_006401963.1	3	unknown
bacterioferritin	*bfr*/Bmul_1081	ABO01605.1	2	cytoplasm

^a^ NCBI accession number of the proteins. ^b^ #matches: number of matching peptides during the identification. ^c^ Subcellular localization prediction for each of the proteins according to Psortdb v3.0.

**Table 3 microorganisms-08-01826-t003:** List of the proteins associated from the biofilm matrix and the OMVs.

Protein	Gene Name/ Identifier	Accession Number ^a^	Subcellular localization ^b^
Porin	*opcP*	WP_054315301.1	Outer membrane
Chaperonin GroEL	*groL*	WP_006400973.1	Cytoplasmic
Citrate (Si)-synthase	Bmul_4643	OJD04157.1	Cytoplasmic
Aldehyde dehydrogenase family protein	Bmul_3451	WP_054317513.1	Cytoplasmic
Malate dehydrogenase	*mdh*	WP_006399543.1	Unknown
Leucine ABC transporter subunit substrate binding protein LivK	Bmul_3623	OJD03740.1	Periplasmic
Glycine dehydrogenase	*gcvP*	GCSP_BURM1	Unknown
Beta-ketoacyl-ACP reductase	Bmul_2181	AOL03371.1	Cytoplasmic
2,3,4,5- tetrahydropyridine-2,6- dicarboxylate N succinyltransferase	*dapD*	DAPD_BURCA	Cytoplasmic
ATP-dependent Clp protease, proteolytic subunit ClpP	*clpP*/Bmul_1349	EED97471.1	Cytoplasmic
YceI family protein	Bmul_2669	EED98206.1	Unknown
Thiol: disulfide interchange protein DsbA/DsbL	*dsbA*	WP_006398112.1	Periplasmic
Redoxin family protein	Bmul_1331	KOS88719.1	Cytoplasmic
Hypothetical protein CA831_04455 partial	Bmul_4929	OXH91981.1	Unknown
Acyl carrier protein	*acpP*	ACP_BURCA	Cytoplasmic
Thioredoxin	Bmul_1445	ABC39250.1	Cytoplasmic
Conserved hypothetical protein	Bmul_0436	EBA48742.1	Unknown
Endoribonuclease L-PSP	Bmul_0927	WP_006398680.1	Unknown
50S ribosomal protein	*rplL*	RL7_BURM1	Cytoplasmic

^a^ NCBI accession number of the proteins. ^b^ Subcellular localization prediction for each of the proteins according to Psortdb v3.0.

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
