# Peer review of "Proteomic Studies of the Biofilm Matrix including Outer Membrane Vesicles of Burkholderia multivorans C1576, a Strain of Clinical Importance for Cystic Fibrosis"

_microorganisms, 2020, doi:10.3390/microorganisms8111826_

Round 1

Reviewer 1 Report

Burkholderia multivorans is currently the most frequently isolated Bcc species from newly colonized CF patients. This manuscript describes the proteome of biofilm matrix and outer membrane vesicles (OMVs) of Burkholderia multivorans C1576 – strain of significant importance in the clinic. The major conclusions is that the most important sources of proteins in biofilm matrix are lysed bacterial cells and OMVs. The authors used also STRING v11 database for performing a protein-protein association network. Therefore a paper is a nice example of joining experimental LC/MS-MS data with bioinformatics. Overall, there is useful data presented. However, there are some issues that need to be clarified. Furthermore, the manuscript in several parts (i.e. the introduction) should have the English language corrected

Specific comments:

  1. Based on the presented work scheme (Figure 1) it seems to be obvious that two proteomes were analyzed and compared, (i) the biofilm matrix proteome containing OMVs and (ii) the proteome of isolated OMVs. Therefore it is not surprising (lane 179-180) that OMV-associated proteins are present in matrix biofilm. The question could be why only 29% not more OMVs proteins is found in matrix?.
  2. In reference to point 1, the compared proteomes should be designated as (i) biofilm matrix containing  OMVs and (ii) OMVs
  3. Why was the moderate confidence level 0.4 used in interaction network analyses?  Did the authors try to do analyses at a higher confidence level such as for ex 0.8-0.9. Would the conclusion be the same in this case?
  4. Figure 3 the matrix lane is overloaded. There is no information about SDS-PAGR percentage, staining etc.
  5. Line 82 – OD=0.13 – how many CFU/ml is it?
  6. Figures 4, 5 6 and 7 are completely fuzzy. Their quality should be improved.
  7. Line 46-47 ….their (i.e OMVs) biological role has not been disclosed yet…– the biological role of OMVs is well known and many recent papers confirm that.

Author Response

1- Based on the presented work scheme (Figure 1) it seems to be obvious that two proteomes were analyzed and compared, (i) the biofilm matrix proteome containing OMVs and (ii) the proteome of isolated OMVs. Therefore it is not surprising (lane 179-180) that OMV-associated proteins are present in matrix biofilm. The question could be why only 29% not more OMVs proteins is found in matrix?.

We thank the reviewer for this comment. Figure 1 was probably misleading and we have now changed and corrected it to clarify better the process of purification and the material analyzed. The text was changed accordingly. As reported in Materials and Methods we analyzed 2 separate fractions, matrix proteins (in the original figure 1 this was wrongly identified as Biofilm), and after separation from the matrix with ultracentrifugation the OMVs. So this procedure accounts for the limited overlap in protein composition between the two samples.

2- In reference to point 1, the compared proteomes should be designated as (i) biofilm matrix containing  OMVs and (ii) OMVs.

As stated above in response to comment 1 the proteomes we are comparing are i) matrix and ii) OMVs. We have corrected Figure 1 for a better understanding as it was confusing.

3- Why was the moderate confidence level 0.4 used in interaction network analyses?  Did the authors try to do analyses at a higher confidence level such as for ex 0.8-0.9. Would the conclusion be the same in this case?

We have used a moderate confidence level for the STRING analysis, as we were searching relations and interactions between these proteins that help us to understand their role. We selected a threshold that allows us to look for the interactions with a sufficient level of confidence (excluding all the false positives) but not that strict to loose possible interesting interactions. We have performed the analysis with the highest level of confidence (0.9) and of course, the interactions are reduced but the strongest interactions were maintained (represent with the thicker line in the original figure). Noteworthy, both networks (matrix and OMVs) have still more interactions than what is expected, this enrichment indicates that the proteins are at least partially biologically connected as a group.

4- Figure 3 the matrix lane is overloaded. There is no information about SDS-PAGEpercentage, staining etc.

We have modified Fig 3 A that now shows two different dilutions for both samples. 

All technical details are now present in the text. Please see section 2.4 of Materials and Methods. Now it says “Samples were separated by SDS-PAGE using a 12% Tris-Glycine polyacrylamide. Stock solutions were diluted 1:2 and 1:5 and 15 µL of both samples were mixed with 6x SDS loading buffer (0.25 M Tris-HCl pH 6.8, 8% SDS, 40% sucrose, 0.5% β-Mercaptoethanol, 0.2% bromophenol blue) and boiled for 5 min. The gel was electrophoresed at 80 V for 5 min and then at 120 V for 60 min in Tris-Glycine SDS buffer and stained with Coomassie Blue.”.”

5- Line 82 – OD=0.13 – how many CFU/ml is it?

Corrected- It corresponds to 1×106 CFU/mL, we have now added in the text.

6- Figures 4, 5 6 and 7 are completely fuzzy. Their quality should be improved.

During the submission process, all figures were uploaded as high quality TIFF, probably they were not available to the referee. We are sorry for this. We have now changed figures to improve quality directly into the final PDF.

7-Line 46-47 ….their (i.e OMVs) biological role has not been disclosed yet…– the biological role of OMVs is well known and many recent papers confirm that.

We thank the reviewer for this comment we have changed it accordingly now it says “OMVs are recognized as biofilm matrix components [10] and their biological role has been related to protective mechanisms [11] as they are involved in a wide range of phenomena like pathogenesis, bacterial communication, bacteria-host interactions, nutrient capture, HGT and competition [12, 13].”

Reviewer 2 Report

In this manuscript, the authors determined the proteome of the biofilm matrix and outer membrane vesicles from a Burkholderia multivorans isolate obtained from an individual with cystic fibrosis.  The proteins identified include those for scavenging reactive oxygen species, iron acquisition, and proteins that protect the bacteria from the host immune system.  The strength of the manuscript is that it provides a snapshot into the proteome of the biofilm matrix and OMVs and the discussion links the data presented to that found in other bacteria.  The weakness is that despite this analysis many proteins of unknown function were identified, thus the biological impact of the proteome requires further elucidation.  The data are essential towards our understanding, but the impact is relatively low.  With that being said, the manuscript is well-written and the data are presented in a coherent format.  

Author Response

We thanks the referee for the support to our work

Reviewer 3 Report

The manuscript of Terán et al. performs the proteomic analysis of proteins associated with the biofilm matrix of B. multivorans C1576, a strain isolated from a CF patient. The authors observe by TEM that B. multivorans C1576 releases OMVs to the biofilm matrix. So, in this work is also the proteomic analysis of the OMVs formed. In summary, the work gives interesting new information about the biofilm matrix protein content and OMVs formed of an important CF pathogen. However, I have some comments and suggestions to the authors for the improvement of the manuscript.

Major comments:

-       Page 3, lines 110-112 and Page 4, lines 149-151: What was the reason for the separation of the proteins fraction by SDS-PAGE, cutting of the bands, purification and then LC-MS/MS analysis? Was because of the biofilm matrix protein sample complexity? These additional steps will increase the loss of some proteins and gel based LC approaches are known to be biased, decreasing the possibility of identification of smaller size proteins and proteins with very low or high pIs. Actually, looking at the Figure 3A (lane Matrix), and on my experience with LC-MS/MS analysis of similar fractions, I was expecting a higher amount of proteins identified in these fraction.

-       Page 5, line 181-184: Please describe in the text the 4 common proteins, because is not clear in the tables this information.

-       Table 1: What is the meaning of “cell” in subcellular localization?

-       Table 1, 2 and 3: A table footnote is missing, similar to the one made in Table 1 (supplementary file). The corresponded “gene name identified” column is also missing in these tables (also present in Table 1 supplementary file). In Figure 6 and 7 and in section discussion is always mentioned the “gene names”, so their introduction in these tables will make the link with the rest of the work.

-       Figure 4 and 5: Please improve the quality of the images, the letters are unfocused.

-       Figure 4A: The letter T appears two times and ND is not described in the legend.

-       Section 3.6, line 269: “The remaining 34 proteins did not show any interaction”. Please confirm if this number is correct, because if they were previously observed interaction for 117 proteins, it’s missing 44 proteins to a total of 161.

-       Figure 7: The Letters from the legend don’t correspond to the figure, they are the same of the legend of the Figure 6.

-       Although limited, exist some manuscripts about OMVs in Burkholderia cepacia complex or Burkholderia genera. I found only 1 article cited. So, I suggest a revision of the OMV literature in the Burkholderia context.

Minor comments:
  • Page 3, line 130-132: Please rephrase the sentence.
  • Results: In this section, several strains are not in italic.
  • Supplementary file Table 1: in the column “Accession number” same lines have red colour. What is the reason?

Author Response

-       Page 3, lines 110-112 and Page 4, lines 149-151: What was the reason for the separation of the proteins fraction by SDS-PAGE, cutting of the bands, purification and then LC-MS/MS analysis? Was because of the biofilm matrix protein sample complexity? These additional steps will increase the loss of some proteins and gel based LC approaches are known to be biased, decreasing the possibility of identification of smaller size proteins and proteins with very low or high pIs. Actually, looking at the Figure 3A (lane Matrix), and on my experience with LC-MS/MS analysis of similar fractions, I was expecting a higher amount of proteins identified in these fraction.

We thank the reviewer for this comment that allows us to better describe our experimental approach.

One of the reasons for the separation by SDS-PAGE was due to the sample complexity. This strain of B. multivorans, the  C1576 produces exopolysaccharides (like cepacian and EpolC1576) that are released to the matrix and that can interfere with protein identification. As polysaccharides are neutral, they are not separated in SDS PAGE, and therefore they are not entering the gel. A second point to consider is that, since we did not adopt the SCX/RP-HPLC bidimensional strategy to analyze the peptides in LC-MS/MS, we decided to adopt a prefractionation method based on SDS-PAGE dividing each lane into five fractions.

We are aware of the fact that by using a prefractionation method there is the possibility to lose some proteins (but partially this is compensated by avoiding the competition between analytes for charge acquisition during the electrospray ionization) and we are also aware of the fact that we are not using the cutting-edge mass spectrometry technology that could strongly improve the sensitivity of our analysis. However, we were in any case able to provide a bird's eye view of these proteomes that is fundamental to lay down the basis for future research in order to provide functional insights into the role of proteins in the matrix and outer membrane vesicles of Burkholderia multivorans. 

-       Page 5, line 181-184: Please describe in the text the 4 common proteins, because is not clear in the tables this information.

The common proteins are now listed in the text and we hope that is clearer now, it says “On the other hand, when considering the twenty most abundant matrix-associated proteins only five of these were shared with the ones found in OMVs: an acyl carrier protein (acpP - ACP_BURCA); a beta-ketoacyl-ACP reductase (Bmul_2181 - AOL03371.1); a thiol: disulfide interchange protein DsbA/DsbL (dsbA - WP_006398112.1); a thioredoxin (Bmul_1445 - ABC39250.1) and an endoribonuclease (Bmul_0927- WP_006398680.1), thus suggesting that OMVs do not represent an important source of proteins for the biofilm matrix of B. multivorans C1576.”

Please see lines 182-185.

-       Table 1: What is the meaning of “cell” in subcellular localization?

It has been changed accordingly into the categories that are used through this paper: cytoplasmic, outer membrane, periplasmic, cytoplasmic membrane, extracellular, unknown.

-       Table 1, 2 and 3: A table footnote is missing, similar to the one made in Table 1 (supplementary file). The corresponded “gene name identified” column is also missing in these tables (also present in Table 1 supplementary file). In Figure 6 and 7 and in section discussion is always mentioned the “gene names”, so their introduction in these tables will make the link with the rest of the work.

We thank the reviewer for this comment we have done it accordingly and we think that is clearer now.

-       Figure 4 and 5: Please improve the quality of the images, the letters are unfocused. 

During the submission process, al figures were uploaded as high-quality TIFF, probably they were not available to the referee. WE are sorry for this inconveninece. We now improved the quality to provide directly a suitable PDF file.

-       Figure 4A: The letter T appears two times and ND is not described in the legend.

Changed accordingly, the repetition of T category has been deleted and the description of ND in the legend has been reported.

-       Section 3.6, line 269: “The remaining 34 proteins did not show any interaction”. Please confirm if this number is correct, because if they were previously observed interaction for 117 proteins, it’s missing 44 proteins to a total of 161. 

We thank the reviewer for the correction, indeed there are 44 proteins that did not show any interaction.

-       Figure 7: The Letters from the legend don’t correspond to the figure, they are the same of the legend of the Figure 6.

Changed accordingly.

-       Although limited, exist some manuscripts about OMVs in Burkholderia cepacia complex or Burkholderia genera. I found only 1 article cited. So, I suggest a revision of the OMV literature in the Burkholderia context.

Indeed, we have added in the introduction references to the literature about the OMVs in Burkholderia, and the text was changed as follows: “The presence of proteins has been recognized in biofilms of many bacterial species, but their identification and functional characterization have been addressed only for a limited number of bacteria [27], including some Burkholderia species [28, 29, 30]. At the same time Burkholderia OMVs have been investigated mainly in the frame of vaccine development [31-33], while there is little information about their proteome [34]”

Also in the discussion (Lines 356-357) a new reference for OMVs sizes in another species of Burkholderia has been added.

Minor comments:

  • Page 3, line 130-132: Please rephrase the sentence.

Done. Now it says: “The hypergeometric distribution was assayed to evaluate the enrichment of the different categories (functional and in terms of subcellular localization) encoded in the genome of B. multivorans C1576 and related to the ones identified in the matrix and in the OMVs.”

  • Results: In this section, several strains are not in italic.

 We apologize for this; it has been corrected accordingly.

  • Supplementary file Table 1: in the column “Accession number” same lines have red colour. What is the reason?

Changed, it is now uniformed all the text into black colour.